# Green-Blue Spaces and Population Density versus COVID-19 Cases and Deaths in Poland

**DOI:** 10.3390/ijerph18126636

**Published:** 2021-06-20

**Authors:** Tadeusz Ciupa, Roman Suligowski

**Affiliations:** Department of Environmental Research and Geo-Information, Institute of Geography and Environmental Sciences, Jan Kochanowski University in Kielce, Uniwersytecka 7, 25-406 Kielce, Poland; tadeusz.ciupa@ujk.edu.pl

**Keywords:** green-blue spaces, population density, COVID-19 cases and deaths, relationships, Poland

## Abstract

In the last year, in connection with the COVID-19 pandemic caused by the SARS-CoV-2 coronavirus, scientific papers have appeared in which the authors are trying to identify factors (including environmental) favoring the spread of this disease. This paper presents the spatial differentiation in the total number of COVID-19 cases and deaths during the full year (March 2020–March 2021) of the SARS-CoV-2 pandemic in Poland versus green-blue spaces (green—i.a. forests, orchards, meadows and pastures, recreational and rest areas, biologically active arable land; blue—lakes and artificial water reservoirs, rivers, ecological areas and internal waters) and population density. The analysis covers 380 counties, including 66 cities. This study used daily reports on the progress of the pandemic in Poland published by the Ministry of Health of the Republic of Poland and unique, detailed data on 24 types of land use available in the Statistics Poland database. Statistical relationships were determined between the above-mentioned environmental variables and the variables characterizing COVID-19 (cases and deaths). Various basic types of regression models were analysed. The optimal model was selected, and the determination coefficient, significance level and the values of the parameters of these relationships, together with the estimation error, were calculated. The obtained results indicated that the higher the number of green-blue spaces in individual counties, the lower the total number of COVID-19 infections and deaths. These relationships were described by logarithmic and homographic models. In turn, an increase in the population density caused an increase in COVID-19 cases and deaths, according to the power model. These results can be used in the current analysis of the spread of the pandemic, including the location of potential outbreaks. In turn, the developed models can be used as a tool in forecasting the development of the pandemic and making decisions about the implementation of preventive measures.

## 1. Introduction

In the last year, there have been many scientific publications and reports devoted to the development of the COVID-19 pandemic across the world, including those describing the factors of its spread [1,2,3]. In this process, apart from clinical factors [4,5], meteorological conditions are of great importance [6,7,8,9]. Among these conditions, the most frequently mentioned are air temperature [10,11,12,13], air humidity [14,15,16] and wind speed [17,18]. Subsequently, sub-sections were separated, relating to two environmental factors (green-blue spaces, population density) with which the number of COVID-19 cases and deaths may be related (Figure 1).

### 1.1. Green-Blue Spaces versus Health

A review of the literature indicates that there are few publications presenting the relationships of green-blue spaces with COVID-19 cases and deaths [19]. This is surprising because earlier publications repeatedly indicated the importance of these spaces in sustainable development [20] and on quality of life [21]. Direct contact with green-blue spaces has many health benefits [22,23,24,25,26,27,28,29]. A measurable effect of interactions with green space is an increase in the mobilisation of the body’s immune system and metabolism [30,31,32] and a reduction in the number of cardiovascular [33,34] and respiratory diseases [35,36]. Green-blue spaces are also used in the prevention of infectious diseases [37] and cancer [38,39,40]. These areas even reduce premature mortality [41,42]. Walks in the forest stimulate the activity of leukocytes—natural killers specialised in combating cells that threaten the body, including aggressive pathogens [43]. Spending time in green-blue spaces also supports better mental health and well-being [44,45,46,47,48,49,50]. The value of green-blue spaces was confirmed by research conducted in Great Britain [51], which showed that spending up to two hours a week in green-blue areas results in significant benefits that improve well-being. It is also interesting that people living in the vicinity of green-blue spaces have fewer symptoms of ill health during acutely stressful situations [52]. These areas also play a role in reducing sources of chronic levels of stress for people [53,54,55]. This is because a large area of green space noticeably reduces sources of stress, such as noise levels [56,57], air dust pollution [41,58,59,60,61] and gases, including carbon dioxide [62,63], which in turn leads to an improvement in air quality conditions in that area. The COVID-19 pandemic has highlighted the importance of green-blue spaces as areas that can also mitigate the negative effects of social isolation on mental health [64,65,66,67].

### 1.2. Population Density versus COVID-19

It is widely believed that SARS-CoV-2 is spreading extremely rapidly in densely populated areas [68] as confirmed by research conducted in areas such as India [69], Turkey [70], and the US [71]. Research by Kodera et al. [72] in Japan showed that population density, which is an indicator of social distance, was even more important than meteorological factors in affecting the spread of the virus. A significant correlation (p < 0.05) between the spread and duration of the first phase of COVID-19 decline and population density in several cities in China, England, Germany and Japan was observed [73]. In turn, a slightly positive correlation between population density and the cumulative number of COVID-19 cases in 1055 regions of the world was confirmed [74]. The impact of population density on the incidence rate is also discussed in the case studies from Iran [14] and Algeria [75]. In addition, studies conducted in 913 US counties [76] did not show statistically significant relationships between population density and the infection rate.

### 1.3. Study Aim and Hypotheses

A review of the literature on this subject shows that to the best of the authors’ knowledge, no studies have been conducted to date in any country in Europa (divided into small administrative units, including cities) showing the relationship between the spatial differentiation in the COVID-19 pandemic development, expressed by infection and death indices, and the size of green-blue space and population density—during the duration of specific pandemic waves or over a longer period of time.

The aim of the research is to show the relationship between the size of green-blue spaces and population density and the spatial differentiation in the number of COVID-19 cases and deaths within the entire territory of Poland (380 counties, including 66 cities), during the full year of the SARS-CoV-2 coronavirus pandemic. 

We hypothesize the following: (H1) a large share of green-blue spaces moderates disease risk of COVID-19; (H2) population density exhibits a positive connection with COVID-19.

## 2. Materials and Methods

### 2.1. Study Area

Poland is located in Europe, where it ranks 9th in terms of total area (a total area of 312.7 thousand km^2^) and 8th in population size (38.4 million inhabitants). It is located between the Baltic Sea and the mountains (Carpathians and Sudetes). It has a varied natural environment (although 90% of the area is lowlands) and land use (predominance of agricultural land—over 60% and a large share of forests—approx. 30%). Along the Baltic coast there is a narrow strip of coastal lowlands, and further south there are lakelands (West Pomerania, Pomerania, Lubuskie, Kujawy-Pomerania, Warmia-Masuria and Wielkopolska Provinces)—with a large number of lakes (over 9,000 with an area of over 1 ha). Then there are the Środkowopolskie Lowlands (Mazovia, Łódź, and Lower Silesia Provinces) and the Polish Uplands. The western part of the second region is occupied by the Silesia province, in which the Upper Silesian agglomeration—based on coal resources, has developed, and the eastern part—by the Lublin Province (with the dominance of agricultural land functioning thanks to fertile soils). To the south of them, there is a belt of pre-mountain valleys (with agricultural lands and industrial centers, including the city of Cracow) and the Karpaty and Sudety Mountains (on the slopes—predominance of forests, and in river valleys—housing estates and agricultural lands).

In terms of the administrative division, there are 16 provinces in Poland and 380 counties in total, of which 66 administrative units are city counties, and the remaining 314 are land counties (Figure 2). The largest city is the capital city of Warsaw (approx. 1.8 million inhabitants). The counties are highly diversified in terms of size (from 13.3 km^2^—Świętochłowice to 2976.4 km^2^—Białostocki) and population density (from 19 people per km^2^—Bieszczadzki up to 3,723 people per km^2^—Świętochłowice) [77].

Green-blue spaces in Poland account for 91.2% of the total area, of which green space accounts for 89.1% and blue space accounts for 2.1%. The country’s forest cover equals 30.9%. There is a large regional spatial differentiation in green-blue spaces within the 380 counties. The variability in the proportion of green-blue spaces ranges from 26.3% (in Chorzów—Silesia Province) to 96.4% (county of Bieszczady—Podkarpackie Province). Of the counties, 232 counties have percentages of green-blue spaces that are above the national average (91.2%). The share of green-blue spaces exceeding 94% of the total area occurs in 35 units located mainly in mountainous areas (Bieszczady) and lake districts (Pomerania and Wielkopolska Provinces). These areas are dominated by forests, which constitute a natural ecosystem that plays an essential role in the process of photosynthesis/assimilation (CO_2_ sequestration) and is the largest absorber of net greenhouse gas emissions [78]. In Poland, in the LULUCF (land use, land-use change and forestry) sector, forests absorb an average of 4.42 tons of CO_2_ per 1 ha per year [79], which correlates to a total of approximately 42.7 million tons of CO_2_ a year. It is especially important when trees grow in the vicinity of the urban, industrialised, built-up and communication areas that are characterised by increased emissions of greenhouse gases. These trees are beneficial for maintaining the liveability of these areas [80,81,82].

The map below (Figure 3a) highlights the administrative units with a low green-blue spaces index (<50%). They consist of 20 city counties, concentrated mainly in Silesia Province (7), which is the most industrialised region of Poland (the Upper Silesian Industrial Region).

The population density in Poland in 2020 was 123 people per 1 km^2^ [83]. The highest population density is observed in highly industrialised areas in the southern part of the country, including Silesia (366) and Małopolska (255) Provinces. In turn, the areas of eastern and northern Poland are characterised by the lowest population density values—Podlaskie (58), Warmia-Masuria (59) and West Pomerania (74). These provinces have a predominance of agriculture and forestry and are minimally industrialised, with a large percentage of the rural population. At the county level, the highest population density occurs in the city counties of Upper Silesia, where the population density exceeds 2000 people per 1 km^2^ in 6 of these city counties, and in Świętochłowice and Chorzów, the population density equals 3723 and 3243 people per 1 km^2^, respectively. The population density is greater than 2000 people per 1 km^2^ in another 12 cities (including Warsaw—3461; Cracow—2384; Łódź—2319; and Poznań—2042 people per 1 km^2^) and in other metropolitan centres, as described [84]. The counties located in the vicinity of the abovementioned units (the largest cities) are also characterised by a high population density exceeding 1,000 people per 1 km^2^ (Figure 3b). The counties with the lowest population density are also highlighted in the spatial image. The population density is the lowest in the following counties: Bieszczadzki—19; Sejny—23, Hajnowski—26, Suwałki—27 people per 1 km^2^ (Figure 3b). In 30 counties, the population density is less than 40 people per 1 km^2^. The regional variations in the occurrence of green-blue spaces and population density clearly correspond with each other (Figure 3a,b) as illustrated by the fact that city counties with a relatively low share of green-blue spaces have high population densities.

### 2.2. Data

Data on COVID-19 new cases and deaths are collected by the Ministry of Health of the Republic of Poland and published in daily reports for counties [83]. The study was based on these data, covering a full year, i.e., from the beginning of the pandemic (since 4 March 2020) to 3 March 2021. Moreover, the data for the areas with green-blue spaces were also included. Data on blue-green spaces are available in the database of the Local Data Bank [85] in the group “Geodetic area (data from the Head Office of Geodesy and Cartography)” and in the sub-group “Geodetic area of the country according to the directions of use”. This database is unique to Europe. It was developed in accordance with the provisions of Polish law on land and building records [86,87] and is the largest collection of this type of information that is used in Poland for statistical purposes. The basic sources of these data are materials and information from national geodetic and cartographic resources as well as the results of photogrammetric and geodetic measurements (these measurements even take into account areas smaller than 0.1 ha). Data were registered in a Microsoft Office Excel 2013 database (Microsoft, Redmond, WA, USA).

### 2.3. Statistical Analysis

Based on the summary of daily statistics on the coronavirus pandemic, the following indices were calculated: the total (yearly) number of COVID-19 cases and deaths per 100 km^2^ in the county. These indices are relative measures that enabled the assessment and comparative analysis of individual units, regardless of their size:(1)$$\mathrm{Index}\;\mathrm{of}\;\mathrm{total}\;\mathrm{number}\;\mathrm{of}\;\mathrm{COVID}-19\;\mathrm{cases}\;(I_{C\left(\mathrm{COVID}-19\right)})=\sum_{1}^{365}\left(\frac{\mathrm{COVID}-19\;\mathrm{cases}\;}{\mathrm{total}\;\mathrm{county}\;\mathrm{area}\;}\right)\cdot 100\;\left[{\mathrm{people}\;\mathrm{per}\;100\;\mathrm{km}}^{2}\right]$$
(2)$$\mathrm{Index}\;\mathrm{of}\;\mathrm{total}\;\mathrm{number}\;\mathrm{of}\;\mathrm{COVID}-19\;\mathrm{deaths}\;(I_{D\left(\mathrm{COVID}-19\right)})=\sum_{1}^{365}\left(\frac{\mathrm{COVID}-19\;\mathrm{deaths}\;}{\mathrm{total}\;\mathrm{county}\;\mathrm{area}\;}\right)\cdot 100\;\left[{\mathrm{people}\;\mathrm{per}\;100\;\mathrm{km}}^{2}\right]$$

Three (3) types of areas were identified in the counties and classified as green, blue and grey spaces. The green spaces consisted of forests, woody and bushy lands, orchards, permanent meadows and pastures, recreational and rest areas (areas of recreation centres, children’s playgrounds, beaches, landscaped parks, squares, lawns, and sports areas) and other areas (abandoned land, flood embankments, and dikes). The green spaces also included arable land, which is biologically active for most of the calendar year. For example, Conteese et al. [88] pointed out that agricultural land provides a feasible opportunity to increase public green space access. Moreover, the authors stated that agriculture may be “a complementary form of green space provision with a distinctive value”. Xie et al. [89] and Ma et al. [90] expressed a similar point of view and described agriculture spaces as a complementary form of urban green space offering not only food provisioning but also important social, cultural, and environmental functions including the creation of the sustainable urban environment. 

The blue spaces included lakes and artificial water reservoirs, rivers, ecological areas (mid-field and mid-forest ponds, oxbow lakes, and swamps) and internal waters. The remaining areas were grey space: built-up and urbanised areas (residential and industrial areas and agriculture built-up areas), roads, railways, open-pit mines and spoil tips. In the process of analysis, the cartographic research method was also used, and thematic maps were developed, showing the proportion of green-blue spaces, population density and the differences in the values of the cases and death indices for COVID-19.

The compilation of this information allowed us to develop comparable, commonly used quantitative indicators showing the differences in the percentages of green, blue and grey spaces. Green and blue spaces were treated together due to their similar, beneficial influence on human health.

Next, the dimensionless green-blue/grey index was calculated, showing the relationship between the space that was favourable from the point of view of living conditions, i.e., the summed area of the green and blue space and the grey space:(3)$$\mathrm{Index}\;\mathrm{of}\;\mathrm{green}-\mathrm{blue}/\mathrm{grey}\;(I_{\mathrm{GB}/G})=\frac{\mathrm{green}\;\mathrm{area}\;\left[{\mathrm{km}}^{2}\right]+\;\mathrm{blue}\;\mathrm{area}\;\left[{\mathrm{km}}^{2}\right]}{\mathrm{grey}\;\mathrm{area}\;\left[{\mathrm{km}}^{2}\right]}\;\left[-\right]$$

Statistical relationships were determined between green-blue spaces index (%), green-blue/grey index (–), population density (people per 1 km^2^) and the index of the total number of COVID-19 cases (people per 100 km^2^) and the index of the total number of COVID-19 deaths (people per 100 km^2^) in particular counties and provinces. Various basic types of models were analysed (i.e., logarithmic, power, homographic, exponential, and linear). Of them, the optimal model was selected, and the quality of their fitting (R^2^), significance level and the values of the parameters of these relationships, together with the estimation error, were determined. The level of statistical significance was set at *p* < 0.01. All statistical analyses were performed using STATISTICA software package (TIBCO Software Inc., version 13, Palo Alto, CA, USA).

In the process of analysis, the cartographic research method was also used, and thematic maps were developed, showing the proportion of green-blue spaces, population density and the differences in the values of the cases and death indices for COVID-19.

## 3. Results

### 3.1. Progress of COVID-19 in Poland

The first case of coronavirus infection from SARS-CoV-2 in Poland was confirmed on 4 March 2020 (Zielona Góra—Lubuskie Province), and the first death was confirmed on 12 March (Poznań). By the end of March 2020, 2,099 infected people (the largest number in Masovian Province—26%) and 25 deaths had been confirmed. Until mid-July 2020, the level of daily infections remained at a stable level of several hundred cases (Figure 4), and deaths at several dozen (Figure 5). Starting from mid-August 2020, more than a two-fold increase in cases and deaths was observed in Poland, starting the second wave of the pandemic (Figure 4). This increase probably resulted from the loosening of restrictions in the summer season and the return of children and young people to schools. Starting in mid-September, there was another, very significant increase in cases (27,800 people a day), which lasted until 7 November 2020, and deaths, which lasted until 25 November (674 deaths per day). In the following weeks (until the end of January 2021), a gradual decrease in recorded COVID-19 cases (approximately 5,000 per day) and deaths (approximately 150 people per day) was reported (Figure 5). From the beginning of February 2021, the development of another (3rd) wave of the pandemic was recorded, showing a constant upward trend. On 3 March 2021, i.e., one year after the beginning of the COVID-19 pandemic in Poland, the total number of cases had reached 1.70 million, and the total number of deaths was 36,000.

An analysis of Figure 4 and Figure 5 shows that both cases and deaths showed a large daily variation related to the weekly work cycle of medical services and the number of COVID-19 tests performed. To eliminate Saturday–Sunday fluctuations, a 7-day simple moving average was used.

### 3.2. Indices of the Total Number of COVID-19 Cases and Deaths

During the first year of the pandemic in Poland (4 March 2020–3 March 2021), the average index of the total number of COVID-19 cases reached 548 people per 100 km^2^, and that of the total number of COVID-19 deaths reached 11.5 deaths per 100 km^2^. In particular counties, a very large spatial differentiation in the abovementioned indicators was observed (Figure 6a,b). The former ranged from 61.7 people per 100 km^2^ (Bieszczady—Podkarpackie Province) to 17,766 people per 100 km^2^ (Warsaw). Low values of this indicator (<200 people per 100 km^2^) were registered in 70 units, and in six (6), the indicator was even lower than 100 people per 100 km^2^ (74.0—Suwałki, 78.0—Sejny, 82.0—Mońki, 82.8—Siemiatycze, 85.5 people per 100 km^2^—Ostrołęka). The highest values of the total number of COVID-19 cases exceeding 8,000 people per 100 km^2^ were recorded in 31 city counties. Among them, there were metropolitan centres (Warsaw, Upper Silesian, Cracow, Tri-City, Łódź, Poznań, and Wrocław), around which a ring formed, including the counties with a high indicator value (>2000 people per 100 km^2^) (Figure 6a).

The value of the total number of COVID-19 deaths in counties varied from 1.84 people per 100 km^2^ (Bieszczady—Podkarpackie Province) to 413 people per 100 km^2^ (Świętochłowice—Silesia Province). In 77 counties, it did not exceed 5 people per 100 km^2^. These counties formed clusters in north-eastern Poland (Podlaskie Province) and north-western Poland (West Pomerania and Lubuskie Provinces) in areas with low population densities and distant from urban agglomerations. In turn, the highest values of this index (exceeding 200 people per 100 km^2^) were recorded in 23 city counties that were mainly medium-sized (Leszno—308, Chorzów—298, and Siedlce—292 people per 100 km^2^) (Figure 6b). The above analysis of the maps shows that there was an exceptionally clear spatial relationship between the proportion of green-blue space and the population density and the rate of cases and deaths caused by COVID-19.

### 3.3. Regression Analysis—Models

The relationships between the environmental features and the characteristics of the COVID-19 pandemic in Poland, presented and observed on the maps, were statistically analysed further. There were strong relationships between the green-blue spaces index, green-blue/grey index and population density and the total number of COVID-19 cases and the total number of COVID-19 deaths in all analysed counties (380) (Figure 7, Table 1).

The relationship between the green-blue spaces index vs. the total number of COVID-19 cases and the total number of COVID-19 deaths was illustrated by a logarithmic model with negative slope values (a), which showed that the value of the total number of COVID-19 cases and the total number of COVID-19 deaths decreased when green-blue spaces index increased (Figure 7); the values of the coefficient of determination (R^2^) of the established relationships were high and were 0.811 and 0.804, respectively, at the significance level of *p* = 0.01 (Table 1).

The optimal model for describing the relationship green-blue/grey index vs. the total number of COVID-19 cases and deaths were homographic functions. The fit was obtained at a level similar to that of the previously discussed relationship, i.e., R^2^ = 0.806 and R^2^ = 0.820 for green-blue/grey index vs. the total number of COVID-19 cases and deaths, respectively. The slope parameter was significant at the level of *p* = 0.05 (Table 1). In addition to the increase in the proportion between the share of green-blue spaces in relation to grey space area, the incidence and number of deaths, expressed by the values of the total number of COVID-19 cases and deaths, clearly decreased. The mutual proportion between these two types of space ranged from 0.35 (Chorzów—Silesia Province) to 26.9 (Bieszczady—Podkarpackie Province). Interestingly, the points representing the city counties were grouped along the left branch of the hyperbola (close to the vertical), and the land counties were grouped along the right branch (close to the horizontal). This result indicates a much greater range of index values characterising cases and deaths in the first group of counties. The border between them was set by the hyperbola vertex, which, in relation to the case rate, had the coordinates green-blue/grey index—3.2 and the total number of COVID-19 cases—2200 people per 100 km^2^. This result indicates that within the areas where green-blue spaces account for less than 76% of the total area, there was a sharp increase in the rate of COVID-19 cases, which is characteristic of city counties. The hyperbola vertex, reflecting the relationship between the green-blue/grey index and the total number of COVID-19 deaths, was characterised by the following coordinates: green-blue/grey index—2.5, the total number of COVID-19 deaths—62 people per 100 km^2^. 

The relationships between the population density and the index of the total number of COVID-19 cases and deaths were very strong (Table 1). The coefficient of determination values calculated for the optimal model described by the power regression were very high, and at the significance level of *p* = 0.01, they were R^2^ = 0.943 (I_C(Covid-19)_ = f(PD)) and R^2^ = 0.877 (I_D(COVID-19)_ = f(PD)) (Table 1). The graphic representation of these relationships is illustrated in Figure 7. The analysis of these graphs shows that as the population density increased, the COVID-19 cases and deaths also increased. The points representing land counties gathered in the lower part of the chart and showed a small range of variability, with both population density (from 19 to 673 people per 1 km^2^) as well as the total number of COVID-19 cases (from 61 to 3,297 people per 100 km^2^) and the total number of COVID-19 deaths (from 1.8 to 71.1 people per 100 km^2^). On the other hand, the points characterising the city counties were situated along the middle and upper part of the regression line, which showed high variability in the values of the analysed features, both in terms of population density (from 207 to 3,723 people per 1 km^2^) and index of the total number of COVID-19 cases (from 1,060 to 17,766 people per 100 km^2^) and the total number of COVID-19 deaths (from 19 to 413 people per 100 km^2^). Similarly, a very high relation was confirmed with the use of linear regression.

An analysis of the relationships of the green-blue spaces index, green-blue/grey index and population density and the total number of COVID-19 cases and deaths was carried out for counties located in specific provinces of Poland. The optimal types of model (identified by color) in addition to the quality of their fit to empirical data, expressed by the value of the determination coefficient R^2^, are presented in Table 2.

The types of models specified here correspond to those established for all of Poland. However, these relationships could be better described by other regression relationships. The coefficients of determination calculated for particular provinces were much higher than their equivalents for Poland. They indicate a better match of the data, both in terms of the assessment of the number of cases as a function of green-blue spaces index (R^2^ = 0.8198 ÷ 0.9973) and green-blue/grey index (R^2^ = 0.8198 ÷ 0.9975), as well as deaths at 0.8461 ÷ 0.9929 and 0.8463 ÷ 0.9919, respectively. All the identified models were statistically significant at *p* = 0.01. A graphic image of the described relationships is presented in Figure 8.

The figure focuses on Silesia Province, which is the most industrialised and populated region in Poland as reflected in the average values of the analysed environmental indices: green-blue spaces and green-blue/grey—the lowest in Poland (85.2% and 5.77, respectively) and the highest population density (370.6 people per 1 km^2^).

## 4. Discussion

### 4.1. COVID-19 in First Year of Pandemic in Poland

The analysis of the pandemic development in the first two months in Poland, in spatial terms (in counties), was presented [91]. The Polish government announced an epidemic threat (13 March), with a total of 62 COVID-19 cases [92]. The epidemic has been occurring in Poland since 20 March 2020 [93]. It is assumed that during this period, Poland experienced the first wave of the pandemic, but unlike in Western European countries (Italy, Spain, Great Britain, and France), it was relatively small [94,95,96]. The beginning of the second wave of the pandemic in Poland (August 2021) was also determined by other authors [97].

### 4.2. Environmental Indices versus COVID-19

Many authors pointed to the importance of the green-blue spaces for sustainable development [20] and quality of life [21]. Contact with green-blue spaces has many health benefits [22,23,24,25,26,27,28,29]. Despite this, there are few publications showing a direct link between these types of space and new COVID-19 cases and deaths. Despite this, there are few publications showing a direct link between these types of areas and new COVID-19 cases and deaths [98]. The analysis was performed in 3049 counties located in the US. The presented results fully correspond to our results obtained for 380 counties in Poland during the first year of the pandemic. They confirm the importance of green areas and their impact on shaping the enhanced immunity of the human body and at the same time contribute to reducing the risk of mortality among people with COVID-19. Similar conclusions describing the relationship between the above-mentioned variables were also presented by Lu et al. [19].

Many researchers have studied the relationship between population density and the development of the coronavirus pandemic (cases and deaths) but mainly in large cities around the world. Very similar, strong relationships (R^2^ = 0.94) between the population density and the number of COVID-19 cases in 81 cities in Turkey in March 2020 were observed [70]. A lower but statistically significant relationship at a level of the coefficient of determination R^2^ = 0.50 was described for 48 cities in Algeria during the first wave of the pandemic (March–June 2020) [75]. A moderate positive correlation of the linear model between the population density and the number of cases and deaths in districts in India was confirmed [69], i.e., R = 0.49 and R = 0.59, respectively. American researchers obtained similar results. Based on data from 351 cities in Massachusetts [71], and in New York City [99], observed that the population density had a statistically significant positive effect on the incidence of this disease. A modest correlation between the population density and the number of infections (R^2^ = 0.394) was recorded in 14 prefectures in Japan [72] and in 1055 regions of the world [74]. It should also be observed that there are also publications in which the authors do not show statistically significant relationships between the abovementioned variables. An example of this is the work focusing on 913 counties in the US [76].

## 5. Conclusions

Environmental characteristics, expressed by indicators determining the proportion of green-blue spaces, as well as the proportion of green-blue space to grey space and population density in 380 counties in Poland, significantly explain total COVID-19 cases and deaths in relation to the standardised area of the unit, within one year of the pandemic. Poland is a country suitable for this analysis because there is a large spatial differentiation in natural and socioeconomic conditions, which is very well documented in existing databases. This type of analysis is also favoured by the existing division of counties that are medium-sized administrative units into two groups, i.e., land counties (314) and city counties (66).

The developed maps show that the large proportion of green-blue spaces in counties is generally related to low numbers of COVID-19 cases and death rates. This result was confirmed by the statistical analyses conducted, which showed a high level of correlation between the above-mentioned environmental variables and those characterising COVID-19. These relations were described by the following models: logarithmic (I_C(COVID-19)_) = f(I_GB_), I_D(COVID-19)_ = f(I_GB_)) and homographic (I_C(COVID-19)_) = f(I_GB/G_), I_D(COVID-19)_ = f(I_GB/G_)) with negative, very high, statistically significant correlations. The very well-matched power model for the relation of population density with the total number of COVID-19 cases and deaths indices, showing the directly proportional relationships between these variables, was also valuable. Interestingly, in the land counties, the rates of COVID-19 cases and deaths were different than those in the city counties. This result confirms the impact of the analysed environmental factors on the progress of the COVID-19 pandemic during its first year in Poland. The discovered relationships, at an even greater level of matching, were also confirmed for particular provinces. The indices describing the proportion of green-blue spaces in a given area as well as population density are important characteristics of the geographic environment related to COVID-19 cases and deaths. At the same time, the population density is also easy to calculate.

This study had several limitations. First, the COVID-19 dataset excluded sick persons subjected to tests, which may lead to a biased survey population because of the underrepresentation of these persons. Second, the total number of COVID cases and deaths in the full year was assumed (4 March 2020–3 March 2021), regardless of the course of the pandemic waves. Third, different scientific publications take into account various categories of green-blue spaces, as well as their interpretations. Despite these limitations, we believe that this study is legitimate and useful for providing a foundation to further hypothesis formation about the spread of the COVID-19 pandemic. 

It should be observed that the analysed environmental factors have affected the quality of life and health of the population in a given county for many years. Their negative effects are felt, especially by city dwellers. Even in the event of a pandemic, these conditions can also affect the number of COVID-19 cases and deaths, therefore they should be subject to further exploration. Future research should also use a more sophisticated statistical methodology to better understand the relationships between environmental factors and COVID-19 cases and deaths.

This article is a response to the rising interest in the sustainable development and management of administrative units in the rank of counties in Poland. The research results have scientific and practical significance due to the increasing need for improving residents’ quality of life through the use of green-blue spaces, especially during the coronavirus pandemic.

In summary, dwellers of counties with a lower amount of green-blue spaces (including cities) may be at a high risk of experiencing an infection with coronavirus and mortality due to COVID-19. Indices obtained green-blue spaces and population density in relation to the total number of COVID-19 cases and deaths and could be a useful tool in analysing the spread of the epidemic. Therefore, they may be helpful in making decisions on the introduction of preventive measures, including the identification of larger outbreaks. Research regarding the regional differentiation of the aforementioned types of spaces may be vital when making decisions introducing restrictions, including lockdowns.

## Figures and Tables

**Figure 1 ijerph-18-06636-f001:**
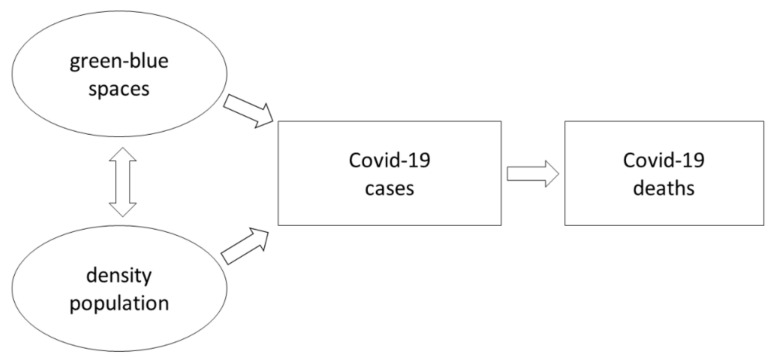
Schematic diagram of research.

**Figure 2 ijerph-18-06636-f002:**
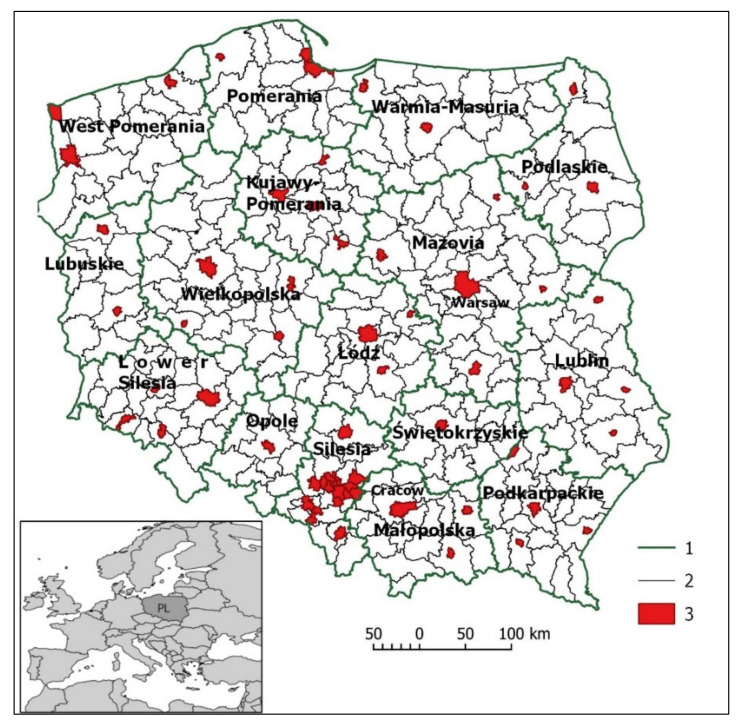
Study area. 1—province border, 2—county border, 3—city counties.

**Figure 3 ijerph-18-06636-f003:**
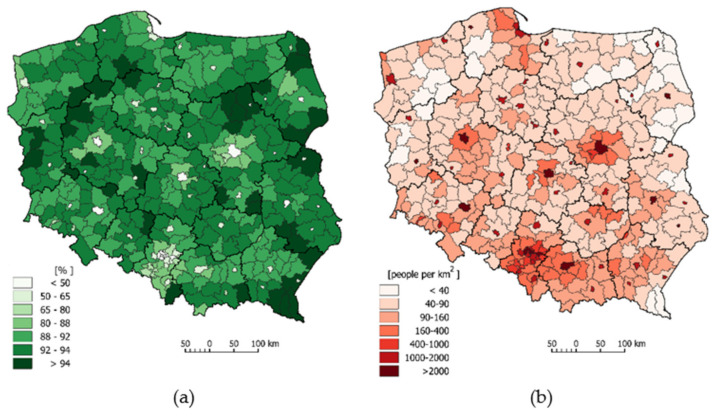
Differences in the values of the indices characterizing: (**a**) green-blue spaces and (**b**) population density in Polish counties.

**Figure 4 ijerph-18-06636-f004:**
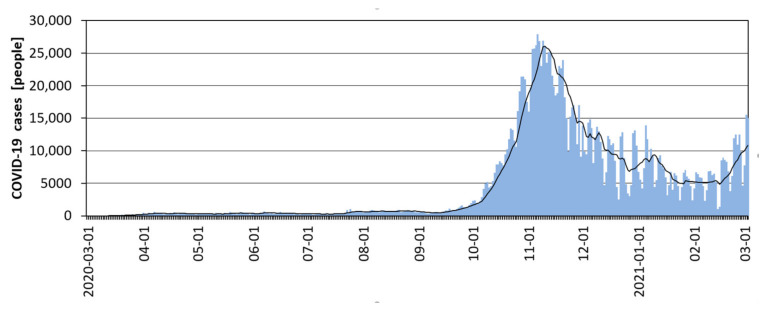
New daily COVID-19 cases in Poland, including the 7-day average, in March 2020–March 2021 (Source: [83]).

**Figure 5 ijerph-18-06636-f005:**
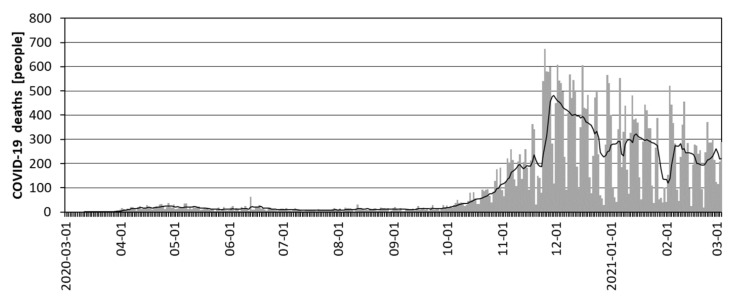
Daily COVID-19 deaths in Poland, including the 7-day average, in March 2020–March 2021 (Source: [83]).

**Figure 6 ijerph-18-06636-f006:**
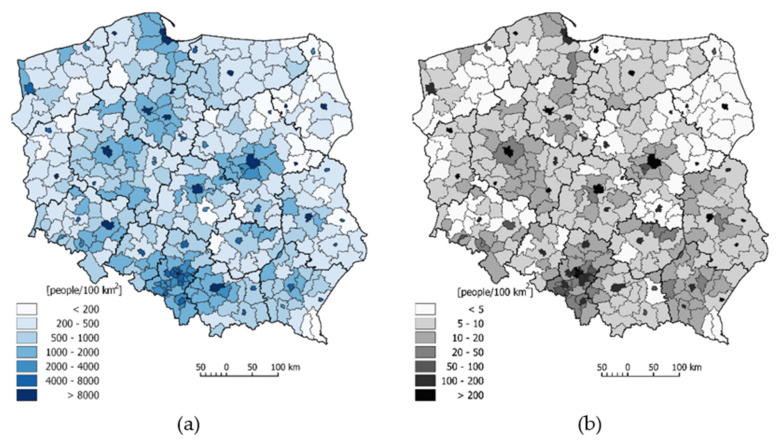
Differences in the values of the following indices of: (**a**) total number of COVID-19 cases; (**b**) total number of COVID-19 deaths in the counties of Poland.

**Figure 7 ijerph-18-06636-f007:**
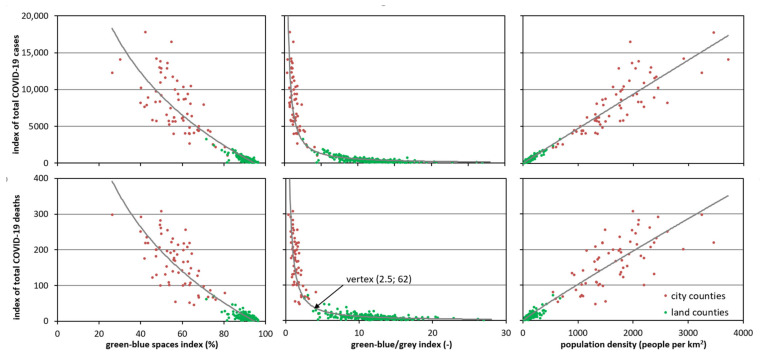
Relationships between the green-blue spaces index (%), green-blue/grey index (–) and population density (people per 1 km^2^) and index of the total number of COVID-19 cases (people per 100 km^2^) and index of the total number of COVID-19 deaths (people per 100 km^2^) in the counties of Poland.

**Figure 8 ijerph-18-06636-f008:**
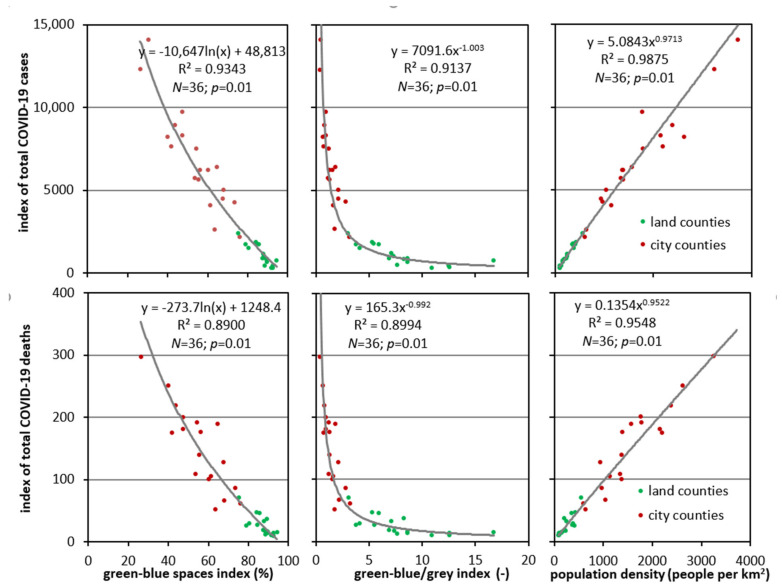
Relationships between the green-blue spaces index (%), green-blue/grey index (−), and population density (people per 1 km^2^) and the indices of: the total number of COVID-19 cases (people per 100 km^2^) and the total number of COVID-19 deaths (people per 100 km^2^) in Silesia Province.

**Table 1 ijerph-18-06636-t001:** Results of the regression analysis between the green-blue spaces index (I_GB_ in %), green-blue/grey index (I_GB/G_, −), and population density (PD in people per 1 km^2^) and index of the total number of COVID-19 cases (I_C(COVID-19)_) and index of the total number of COVID-19 deaths (I_D(COVID-19)_) in the counties of Poland (*N* = 380); in relation to Figure 7.

Index	Formula	R^2^	Parameter	Value	SE	t-Statistic (df = 378)	*p*-Value	Lower 95% CI	Upper 95% CI
I_C(COVID-19)_	a·ln(I_GB_) + b	0.811	a (slope)	−1,653	289.61	−40.238	<0.0001	−12,223	−11,084
			b (const.)	53,076	1283.1	41.367	<0.0001	50,553	55,599
I_D(COVID-19)_		0.804	a (slope)	−226.69	5.7638	−39.329	<0.0001	−238.02	−215.36
			b (const.)	1033.1	25.535	40.456	<0.0001	982.8	1083.3
I_C(COVID-19)_	$\frac{a}{I_{GB/G}}+b$	0.806	a (slope)	7452.3	188.24	39.587	<0.0001	7082.2	7822.4
			b (const.)	−160.91	76.121	−2.1139	0.0351	−310.59	−11.241
I_D(COVID-19)_		0.820	a (slope)	146.86	3.5428	41.454	<0.0001	139.90	153.83
			b (const.)	−2.9567	1.4326	−2.0638	0.0397	−5.7736	−0.1398
I_C(COVID-19)_	a·PD^b^	0.943	a (slope)	4.9766	0.8099	6.1440	<0.0001	3.3839	6.5692
			b (const.)	0.9735	0.0214	45.378	<0.0001	0.9313	1.0156
I_D(COVID-19)_		0.877	a (slope)	0.1618	0.0353	4.5771	<0.0001	0.0923	0.2313
			b (const.)	0.9028	0.0289	31.150	<0.0001	0.8458	0.9598

**Table 2 ijerph-18-06636-t002:** Determination coefficients (R^2^) of the relationships between the environmental characteristics (green-blue spaces index—I_GB_, population density—PD, and green-blue/grey index—I_GB/G_) and the indices of: the total number of COVID-19 cases (I_C(COVID-19_) and the total number of COVID-19 deaths (I_D(COVID-19_) by province.

Province (Number of Counties)	I_C(COVID-19)_ = f(I_GB_)	I_D(COVID-19)_ = f(I_GB_)	I_C(COVID-19)_ = f(I_GB/G_)	I_D(COVID-19)_ = f(I_GB/G_)	I_C(COVID-19)_ = f(PD)	I_D(COVID-19)_ = f(PD)
Lower Silesia (30)	0.8828	0.9286	0.9020	0.9251	0.9703	0.8842
Kujawy-Pomerania (23)	0.9578	0.9796	0.9652	0.9755	0.9742	0.9960
Lublin (24)	0.9128	0.9725	0.8956	0.9563	0.9793	0.9974
Lubuskie (14)	0.9942	0.9929	0.9975	0.9895	0.9735	0.9526
Łódź (24)	0.8787	0.8848	0.8978	0.8763	0.9666	0.9601
Małopolska (22)	0.8712	0.8461	0.8902	0.8463	0.9796	0.9458
Mazovia (42)	0.8198	0.9317	0.9102	0.9415	0.9731	0.9238
Opole (12)	0.9910	0.9867	0.9948	0.9919	0.9997	0.9964
Podkarpackie (25)	0.8773	0.9079	0.8842	0.9096	0.9753	0.9608
Podlaskie (17)	0.9497	0.9379	0.9645	0.9426	0.9868	0.9768
Pomerania (20)	0.9768	0.9754	0.9709	0.9824	0.9909	0.9679
Silesia (36)	0.9343	0.8900	0.9137	0.8994	0.9875	0.9548
Świętokrzyskie (14)	0.9973	0.9697	0.9975	0.9672	0.9990	0.9956
Warmia-Masuria (21)	0.9620	0.9499	0.9744	0.9377	0.9738	0.9530
Wielkopolska (35)	0.9176	0.9457	0.9423	0.9510	0.9514	0.9860
West Pomerania (21)	0.9309	0.9389	0.9523	0.9565	0.9818	0.9869

The colors indicate the optimal type of regression relationship: blue—logarithmic, green—power, orange—homographic, yellow—exponential, grey—linear.

## Data Availability

Data on COVID-19 new cases and deaths are openly available on the following URL: https://dane-i-analizy.pl/raport (accessed on 4 June 2021). Data on blue-green spaces and population density are available in the database of the Local Data Bank on the following URL: https://bdl.stat.gov.pl/BDLS (accessed on 4 June 2021).

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
