# Peer review of "Green-Blue Spaces and Population Density versus COVID-19 Cases and Deaths in Poland"

_ijerph, 2021, doi:10.3390/ijerph18126636_

Round 1

Reviewer 1 Report

Dear editors of IJERPH,

thank you so much for the possibility to review for the journal. I am summarizing my comments below:

- The topic is truly interesting and topically fits to the scope of the journal.

- Abstract. Could you please attempt rewriting the abstract in a more layman style so that it could attract a wider audience of the readers? I would propose to start the abstract with an opening sentence that will give us a bit of background before sharing the objective. I would also avoid the usage of abbreviations in this part, it is not necessary as the abstract and full-text is usually read separately. Moreover, the usage of abbreviations tends to cause confusions so I would personally avoid them. It would be good to identify “a full year” (line 10). Is it please possible to be more specific about the methodology applied?

- The keywords should be more developed to reflect the content and make the search in the databases easier. Green-blue space (or spaces?).

- Introduction serves here also as a theoretical framing of the study. This part is indeed nicely written, however, I think that could be more in-depth. I would suggest dividing this part into several subsections and to focus on the topics independently (at least the conceptualisation of green-blues spaces, population density and its relation to diseases and COVID-19 issues could be separated). I would reduce the usage of abbreviations if it is possible (line 41). It might be good if several hypotheses are defined based on the literature analysis. Is there any possibility to graphically show in a scheme relations between the two above-mentioned topics? This might produce very interesting insights.

- Section 2. Could you please separate the section on Data and the section on Study area description? Data are quite well described but I think that my space should be devoted to the description of the study area. I think we need to know more about national/regional contexts in Poland so that the findings could be better understood and interpretations comprehended.

- It seems to me that Figure 1 is not necessary as it doesn´t bring relevant information.

- In my opinion, the methodology applied should be more in-depth explained (2.2). A graphical scheme showing the steps done would be good here.

- Figure 4 and others: Please consider using a full name of indicator instead of coding (abbreviation)  in the key to maps.

- Results. It seems to me that this part includes not only original results (which is great) but also methodological remarks and other comments. These should be rather shifted elsewhere to make this part more focused.

- Title of the figure 7 – please use full names of indicators.

- To be honest, I don´t like much the style of referencing in the paper. In the majority of refs, the name of authors are avoided which makes the text not so nice readable.

- The discussion should be more developed. Please follow the structure of the results section.

- The conclusion is good, however, more could be said on i) a practical usability and recommendations derived, ii) limitations of the study (methods, data, a period of the study, etc.), iii) theoretical findings could be better implemented in this part.

I think that this is going to be a very nice paper when revised. My suggestion to the authors is to work more on the paper so that its quality could be improved. I hope the authors will find my observations useful. I recommend a major revision.

Many thanks.

Kind regards,

Author Response

Dear Reviewer,

Thank you very much for your insightful remarks and comments. We believe that thanks to them the revised manuscript has become much better and more comprehensible. Our responses to the comments can be found in the attached word file.

Yours sincerely,

Tadeusz Ciupa i Roman Suligowski

Reviewer 2 Report

This is well written and presented paper that adds to the emerging body of literature on the role of urban density and greenspaces during the COVID-19 pandemic. The authors provide a straightforward and convincing case that correlates the occurrence of COVID-19 (both deaths and infections) with green spaces across all of Poland. The strength of the paper is that it provides a macro view of this relationship and that it confirms, on a national level, what is intuitively to be expected. When published, it will be a paper that I will certainly cite.

The following is a suggestion that should not detract from the value of the paper as it is.

Looking at the graph of COVID-19cases in Poland (Figure 2) it would be intriguing to compare the spatial distribution of the early cases say until 1 October, with the peak (1 October 2020 to 31 March 2021 to see how the greenspace/population density plays out in the early period.

Further, if the authors feel inclined, it might be interesting to replicate the Silesia example in a country that is primarily agricultural. However, this may be something for the authors to explore in another paper.

I have one a very minor corrective comment: The vertical axes in figures 6 and 7 are too small to be legible, as are the horizontal axes of figure 2 and figure 3

Author Response

Dear Reviewer,

Thank you very much for your insightful remarks and comments. Our responses can be found in the attached Word file.

Yours sincerely,

Tadeusz Ciupa i Roman Suligowski

Round 2

Reviewer 1 Report

I have no further comments as all my observations on the original version were sufficiently commented and implemented as it was appropriate.

Let me congratulate the authors for a nice paper and wish them all the best for their future work.

It was my pleasure to collaborate on the development of this paper.

Kind regards,